# A Systematic Review and Meta-Analysis on the Real-World Effectiveness of COVID-19 Vaccines against Infection, Symptomatic and Severe COVID-19 Disease Caused by the Omicron Variant (B.1.1.529)

**DOI:** 10.3390/vaccines11020224

**Published:** 2023-01-19

**Authors:** Hassen Mohammed, Dan Duy Pham-Tran, Zi Yi Michelle Yeoh, Bing Wang, Mark McMillan, Prabha H. Andraweera, Helen S. Marshall

**Affiliations:** 1Vaccinology and Immunology Research Trials Unit, Women’s and Children’s Health Network, Adelaide, SA 5006, Australia; 2Robinson Research Institute and Adelaide Medical School, the University of Adelaide, Adelaide, SA 5006, Australia

**Keywords:** COVID-19, Omicron, vaccine effectiveness, COVID-19 vaccines, booster vaccination

## Abstract

Real-world data on the effectiveness of COVID-19 vaccines against the Omicron variant (B.1.1.529) is limited. This systematic review aimed to investigate the real-world effectiveness and durability of protection conferred by primary course and booster vaccines against confirmed Omicron infection, and severe outcomes. We systematically searched literature up to 1 August 2022. Meta-analysis was performed with the DerSimonian-Laird random-effects model to estimate the pooled vaccine effectiveness (VE). Overall, 28 studies were included representing 11 million individuals. The pooled VE against Omicron infection was 20.4% (95%CI: 12.1–28.7%) and 23.4% (95%CI: 13.5–33.3%) against symptomatic infection with variation based on vaccine type and age groups. VE sharply declined from 28.1% (95%CI: 19.1–37.1%) at three months to 3.9% (95%CI: −24.8–32.7%) at six months. Similar trends were observed for symptomatic Omicron infection. A booster dose restored protection against Omicron infection up to 51.1% (95%CI: 43.8–58.3%) and 57.3% (95%CI: 54.0–60.5%) against symptomatic infection within three months; however, this waned to 32.8% (95%CI: 16.8–48.7%) within six months. VE against severe Omicron infection following the primary course was 63.6% (95%CI: 57.5–69.7%) at three months, decreased to 49% (95%CI: 35.7–63.4%) within six months, and increased to 86% after the first or second booster dose.

## 1. Introduction

The Omicron variant (B.1.1.529) was reported to the World Health Organization (WHO) from South Africa in late November 2021 [1]. It was immediately designated as a variant of concern (VOC) [1]. Compared to pre-omicron variants, a number of mutations have been identified in the omicron variant, including multiple mutations in the receptor-binding domain of the spike protein associated with increased transmissibility and immune evasion after natural infection and vaccination [2,3,4]. The omicron variant has rapidly evolved into new sub-lineages or sub-variants: BA.1 (B.1.1.529.1), BA.2 (B.1.1.529.2), BA.3 (B.1.1.529.3), BA.4 (B.1.1.529.4), and BA.5 (B.1.1.529.5). As of 29 November 2022, BA.5, BA.2.75, BA.4.6 and XBB (a hybrid of two different Omicron BA.2 sub-variants) are Omicron sub-lineages being monitored by the WHO to investigate if these lineages may pose an additional threat to global public health [1].

Several studies have shown diminished neutralization of both Omicron variants by pre-Omicron convalescent sera and by sera of vaccinated individuals [4,5,6]. Recent studies have shown a reduction in COVID-19 vaccine effectiveness against the Omicron variant [7,8,9,10,11], affecting the current COVID-19 vaccination strategy. Recent social media analysis has shown increased public vaccine hesitancy due to the potential lack of effectiveness of ancestral COVID-19 vaccines against the new VOCs [12].

Emerging data on high prevalence of asymptomatic infection, greater risk of reinfection, and reduced vaccine protection during the omicron-dominant period compared to the earlier VOC warrants further investigation on the effectiveness of current COVID-19 vaccines against the Omicron variant [7,8,9,10,11,13]. A systematic review and meta-analysis have recently been published to evaluate the effectiveness of the current COVID-19 vaccines against Omicron infection [14]. This meta-analysis included 15 studies and demonstrated that primary vaccination does not provide sufficient protection against symptomatic Omicron infection [14]. However, the systematic review included studies conducted in the early Omicron era with shorter-term follow up. The real-world long-term effectiveness and durability of protection conferred by primary COVID-19 vaccination course and booster doses against the Omicron variant is not precisely known. To summarize the existing evidence on the effectiveness and the duration of protection conferred by COVID-19 vaccines, data were synthesized from an ongoing systematic review [15]. This systematic review aimed to investigate the real-word effectiveness of primary and booster vaccination against SARS-CoV-2 infection and severe COVID-19 disease due to laboratory-confirmed SARS-CoV-2 Omicron variant. The review also aimed to evaluate the duration of protection following full vaccination and booster doses.

## 2. Methods

The systematic review drew data from an ongoing systematic review that aimed to synthesis and evaluate the vaccine effectiveness (VE) of COVID-19 vaccines at preventing SARS-CoV-2 infections and severe COVID-19 disease in real-world settings. The systematic review followed the Preferred Reporting Items for Systematic Review and Meta-Analysis (PRISMA) guideline and was registered in the Prospective Register of Systematic Reviews (PROSPERO registration ID: CRD42022291375) [15].

### 2.1. Search Strategy

Systematic literature searches were performed on MEDLINE, PubMed, Embase and Cochrane Database of Systematic Reviews website on or before 1 August 2022 with no language restrictions. WHO COVID-19 DATABASE, pre-print severs (medRxiv, bioRxiv) and grey literature were searched. For preprint studies with several versions available, the most recent update published up to 1 August 2022 was included. Reviews and their references are examined for inclusion. Medical subject headings with the following search keywords were used: SARS-CoV-2 OR COVID-19 AND Vaccine OR Vaccination AND OR Vaccine effectiveness. The methods have been previously described in detail in the ongoing systematic review protocol [15] and full search strategies are available in Appendix A.

### 2.2. Study Selection

Observational (non-randomized) studies including cohort studies (prospective or retrospective), cross-sectional studies, case control including test-negative design (TND), regression discontinuity design studies, post-licensure observational studies that investigated the effectiveness of COVID-19 vaccines against documented, symptomatic, severe COVID-19 disease (defined as hospitalization, ICU admission, intubation or mechanical ventilation, or death) were included. COVID-19 cases were defined as being due to the Omicron variant infection, based on S target–negative results on PCR or whole-genome sequencing. VE studies that evaluated effectiveness of the primary vaccine course and booster vaccination compared to no vaccination were included. Outcomes of interest were VE against Omicron infection of “any type” (i.e., studies did not indicate underlying symptoms), “symptomatic COVID-19”, and “severe COVID-19” due to Omicron infection. We only included studies that evaluated VE ≥14 days after the primary vaccination course, and ≥7 days after the booster dose. No restrictions were applied to the age of participants, the types of vaccination, or the number of participants. Heterologous primary schedules were considered. Studies that did not report VE data or did not use any confounder adjustment strategies were excluded.

All the relevant records were screened by title and abstract. The retrieval results were screened with the help of Endnote and duplicate studies were eliminated. Potentially relevant publications underwent full-text examination and disagreements on eligibility were solved through discussion. The full texts suitable for the quantitative synthesis were collected in an excel spreadsheet for data extraction.

### 2.3. Data Extraction

Data were extracted by three (H.M., D.D.P.-T. and Z.Y.M.Y.) independent reviewers to identify eligible studies that met pre-specified inclusion criteria. The following information: study design, year of publication, country, age, population type, type of vaccines, time period post primary series or booster doses and study follow-up period, were extracted from the eligible studies. VE data were stratified according to SARS-CoV-2 vaccination course, ≥14 days after completion of the primary vaccination course and ≥7 days after receiving the first or second booster doses. Within each subgroup, the vaccination course was classified according to the vaccine type or technology. Duration of effectiveness of SARS-CoV-2 vaccines was assessed in intervals of three, six, and longer than six months after the primary vaccination series, whereas for the booster vaccination, shorter time intervals were considered (seven or more days, within three months, three to six months) due to less follow-up time since introduction.

### 2.4. Quality Assessment

The Joanna Briggs Institute (JBI) tools [16] were used to assess risk of bias of the included studies (Appendix A).

### 2.5. Data Analysis

Descriptive statistics were used to summarize the characteristics of studies included in this review. VE was quantified as the risk reduction of any or severe Omicron infection, expressed as a percentage, compared to the unvaccinated group. VE estimates were derived from regression models (Logit, Poisson, and Cox regression models) and calculated as (1 − IRR) × 100, where IRR = incidence rate ratio; (1 − HR) × 100, where HR = Hazard ratio; (1 − RR) × 100, where RR = Risk ratio; and (1 − OR) × 100, where OR = Odds ratio is the ratio of the rate of COVID-19 in the vaccinated group to the corresponding rate in the unvaccinated group. The DerSimonian-Laird random-effects model with Hartung- Knapp-Sidik-Jonkman variance correction was used to combine VE estimates. We used the *I*^2^ test to quantify the heterogeneity between studies. *I*^2^ values were defined as low (≤50%), moderate (50–75%), or high heterogeneity (>75%). To estimate the duration of protection following the primary vaccination series, we modeled days since completing the primary course as a continuous effect, allowing for nonlinearity by using restricted cubic splines. The analysis was carried out using Stata 17.

## 3. Results

### 3.1. Characteristics of Studies

The initial search generated 13,601 studies. After removing duplicates, screening titles and abstracts of 7710 potential studies, 909 studies were identified for full text review (Figure 1). After a full text review, 299 studies were included for the ongoing systematic review as of 1 August 2022. In total, 28 observational VE studies on confirmed Omicron cases (22 case-control [9,10,11,17,18,19,20,21,22,23,24,25,26,27,28,29,30,31,32,33,34,35] and six cohort studies [7,8,36,37,38,39]) were included in this meta-analysis. Overall, 17 studies [17,18,19,20,21,22,24,25,26,27,28,29,31,34,35,38,39] investigated VE of both primary and booster vaccination, nine primary vaccination course without boosters [7,8,9,11,23,32,36] and two evaluated VE of booster doses only [30,37]. A total of 238 different VE estimates against documented Omicron infection (*n* = 47), symptomatic (*n* = 95) and severe COVID-19 disease (*n* = 96) over different time periods were included in this meta-analysis. Of these, 47% (*n* = 112) were for primary vaccination series and 52.9% (*n* = 126) for booster doses. Of the total 238 VE estimates, 196 (82.2%) were calculated from odds ratio (OR), 29 (12.1%) from the hazard ratio (HR), 10 (4.2%) from incidence rate ratio (IRR) and three (1.2%) from risk ratio (RR).

The included studies had a total of sample size of 11.6 million individuals, and ranged from 1052 to 2,706,008 participants. The majority of the studies were carried out in USA (*n* = 12) [9,10,17,26,28,29,30,31,35,36,37,39] followed by the UK (*n* = 6) [11,25,32,33,34,38], Canada (*n* = 4) [19,20,21,24], Qatar (*n* = 2) [18,22], France (*n* = 1) [27], Denmark (*n* = 1) [7], South Africa (*n* = 1) [23] and Chile (*n* = 1) [8] (Table 1). The majority of the included studies (*n* = 18) evaluated VE in adults ≥18 years of age, while two were exclusively on the pediatric populations (5–12 years of age) [22,23], two on adolescents (12–17 years of age) [11,19], one middle age adults (40–64 years of age) [25] and two on older adults (≥60 years of age) [7,24]. Two studies [37] included high risk populations (patients under hemodialysis therapy). The primary vaccination course represents two doses of BNT162b2 (Pfizer–BioNTech), mRNA-1273 vaccines (Moderna), AZD1222 (ChAdOx1 nCoV-19, AstraZeneca), CoronaVac (Sinovac), or one dose of Ad26.COV2.S (Janssen) vaccines. Most booster studies investigated VE of a single mRNA-based booster dose following the primary vaccination course, and only one study reported VE of a third dose of the vector-based vaccine, AZD1222 [25]. One study evaluated the VE of a second booster dose of the mRNA-1273 vaccines following three doses of mRNA-based vaccines [24] (Table 1). The median follow-up period for booster dose was shorter (16.5 weeks, IQR: 8.5–24) compared to the primary vaccine course (44 weeks, IQR 24–48). The quality assessment of each study using the JBI critical appraisal checklist is listed in Appendix A. The majority of included VE studies accounted for key potential confounders that may have influenced both the receipt of COVID-19 vaccine and the occurrence of SARS-CoV-2 infection. The list of covariates used in final analyses of vaccine effectiveness (VE) estimates from the included primary studies is reported in Appendix A.

### 3.2. Vaccine Effectiveness

#### 3.2.1. VE against Any Omicron Infection following Completion of the Primary Vaccination Course

A total of four studies (24 different VE estimates) [7,8,10,36] were included in the meta-analysis of primary vaccination series against Omicron infection. Two studies were conducted in the USA [10,36], one in Denmark [7] and one in Chile [8]. For all ages and vaccines, the pooled VE against any SARS-CoV-2 Omicron infection was 20.4% (95%CI: 12.1–28.7%, *I*^2^ = 96.4%). The pooled VE estimate within 3 months following the primary vaccination course was 28.1% (95 CI: 19.1–37.1%). The VE estimates varied based on vaccine type, for mRNA based; BNT162b2 38.1% (95%CI: 23.3–52.8%), mRNA-173, 27.8% (95%CI: 6.6–49.0%), BNT162b2 or mRNA-1273, 22.3 (95%CI: 8.6–36.0%) and for vector based; AZD1222, 9.4% (95%CI: −15.5–34.4%). The estimated VE for CoronaVac was 37.8% (95% CI: 36.1–39.6%, data from a single study) in children 3–5 years of age (Figure 2). As displayed on the scatter plot, VE against Omicron infection declined sharply approximately after 2 months (50–60 days) following the primary vaccination course (Appendix A). The pooled VE estimates decreased to 4.0% (95%CI −24.8–32.7%) within three to six months (only mRNA-based vaccines included). There were not enough time data points to estimate pooled VE against Omicron infection after six months (mRNA-1273; 5.9%, 95%CI: 0.60–11.2%, data from a single study) (Figure 2).

#### 3.2.2. VE against Symptomatic Omicron Infection following Completion of the Primary Vaccination Course

A total of 10 studies (39 different VE estimates) were included in the meta-analysis of primary vaccination series against symptomatic Omicron infection conducted in the UK (*n* = 3) [11,25,32], Canada (*n* = 3) [19,20], Qatar (*n* = 2) [18,22], USA (*n* = 1) [9] and France (*n* = 1) [27]. The pooled VE estimate against symptomatic Omicron infection for all ages and vaccine types was 23.4% (95%CI: 13.5–33.3%, *I*^2^ = 99.6%). The pooled VE against symptomatic Omicron disease was 37.1% (95%CI: 26.9–47.2%) after the first three months, declining to 10.6% (95%CI: 4.6–16.5%) between three to six months to −4.3% (95%CI: −15.4–6.7%) after six months (Figure 3). VE against symptomatic Omicron infection had a similar reduction over time, rapidly declining around 40 days following the primary vaccination course, as shown on the scatter plot (Figure 4).

#### 3.2.3. VE against Severe Omicron Infection following Completion of the Primary Vaccination Course

A total of 12 studies (49 different VE estimates) conducted in USA (*n* = 4) [17,28,29,31], Canada (*n* = 3) [19,20,24], Qatar (*n* = 2) [18,22], France (*n* = 1) [27], South Africa (*n* = 1) [23] and Chile (*n* = 1) [8] evaluated VE against severe Omicron infection. Severe COVID-19 was a composite outcome of hospitalization (59.1%), emergency department (14.2%), intubation or mechanical ventilation (14.2%), admission to the intensive care unit (ICU) (6.1%), or death (6.1%). Only two [27,35] studies evaluated the effectiveness of COVID-19 vaccines against death. Two studies [18,24] included death or hospitalization as a composite outcome of severe Omicron infection. Therefore, VE estimates against death were not pooled separately due to the low number of studies and insufficient data. The pooled VE against severe COVID-19 was 56.9% (95%CI: 51.4–62.5%, *I*^2^ = 84.4%) (Figure 5). VE against severe Omicron infection decreased from 63.6% (95%CI: 57.5–69.7%) at three months to 48.3% (95%CI: 39.0–57.6%) at six months following the primary vaccination series (Figure 5). VE remained stable after six months at 49.7% (95%CI: 35.7–63.7%), also shown on the scatter plot (Appendix A). The highest pooled VE estimates against severe Omicron disease were observed following two doses of mRNA-1273 within 3 months (72.5%, 95%CI: 55.7–89.3%, *I*^2^ = 59.7).

#### 3.2.4. VE against Omicron Infection after the First Booster Dose

A total of eight studies conducted in the USA (*n* = 4) [31,35,37,39], Canada (*n* = 2) [21,24] and UK (*n* = 2) [34,38] evaluated VE (17 different VE estimates) [34,38] against Omicron infection. The overall pooled VE estimate of the first booster dose against Omicron infection was 48.8% (95% CI: 42.0–55.6%, *I*^2^ = 97.3%) (Figure 6). The pooled VE estimates of the first booster dose after 7–14 days was 41.4% (95%CI: 32.4–503%, *I*^2^ = 79.8%) and increased to 51.2% (95%CI: 43.9–58.6%) within three months following three doses of any mRNA vaccines. Data from a single study during the same time period showed a slightly lower VE estimates of 47% (95%CI: 13–81%) following two dose adenovirus vector vaccines with one mRNA vaccine (Figure 6). There was insufficient follow-up period post three months to evaluate the waning of booster vaccination against Omicron infection.

#### 3.2.5. VE against Symptomatic Infection after the First Booster Dose

A total of eight studies conducted across Canada (*n* = 4) [19,20,21,24], UK (*n* = 2) [25,33], France (*n* = 1) [27] and Qatar (*n* = 2) [18,22] evaluated VE of one booster dose (50 different VE estimates). The pooled VE of a booster dose against symptomatic infection was 55.9% (95%CI: 53.4–58.4%, *I*^2^ = 98.4) (Figure 7). The VE varied by vaccine types and technology. The pooled VE estimates of three doses of mRNA vaccine after 7–14 days was 58.4% (95%CI: 54.8–62.0%, *I*^2^ = 83.7%). VE of the first booster dose remained stable up to three months following three doses of any mRNA vaccines (56.1%, 95%CI: 50.9–61.3%) or three doses of adenovirus vector vaccines (60.2%, 95%CI: 55.0–65.5%). Slightly lower VE estimates were observed for heterologous boosting using 2-dose adenovirus vector vaccines with one mRNA vaccine (51.7%, 95%CI: 50.9–61.3%). Between the periods three to six months, pooled VE estimates reduced to 32.8% (95%CI: 16.9–48.8%), waning less pronounced for three-dose, homologous mRNA vaccination (Figure 7).

#### 3.2.6. VE against Severe Omicron Infection after the First Booster Dose

A total of 11 studies (39 different VE estimates) conducted in the USA (*n* = 5) [17,26,28,29,30], Canada (*n* = 3) [20,21,24], Qatar (*n* = 2) [18,22] and France (*n* = 1) [27] evaluated VE against severe Omicron infection. The pooled VE estimate of the first booster dose against severe Omicron disease was 86.5% (95%CI: 84.4–88.6%, *I*^2^ = 90.0%). The pooled VE estimates at 7–14 days was 88.1% (95%CI: 85.0–91.2%), 85.1% (95%CI: 80.2–90.1%) within 3 months and 88.0% (95%CI: 80.7–91.2%) within three to six months (three doses of any mRNA vaccines representing most of the pooled VE estimates) (Figure 8).

#### 3.2.7. VE of a Second Booster or Fourth Dose

One study evaluated the VE of a fourth dose (second booster dose) of mRNA-1273 in older residents of long term care facilities in Ontario, Canada [24]. The pooled VE estimate of a fourth dose of mRNA-1273 followed by any combination of three mRNA vaccines at “≈7 days” against Omicron infection was 50.3% (95%CI: 47.1–53.6%), 69.7% (95%CI: 65.3–74.2%) against symptomatic Omicron infection, and 86% (95%CI: 81–90%) against severe outcomes (Appendix A). There was no VE data for other time periods.

### 3.3. Subgroup Analyses

Subgroup VE analyses were performed by age, and sub-lineage of Omicron. For some subgroup analyses, there were not enough data points to estimate VE or duration of protection at different time points.

### 3.4. VE by Age Groups

#### 3.4.1. Pediatric population

The pooled VE estimate for primary vaccination series (CoronaVac or BNT162b2) against Omicron infection in children under 12 years of age was 37.8% (95%CI: 36.0–39.5%) (Appendix A).

#### 3.4.2. Adolescents

The pooled VE against Omicron infection following a two-dose BNT162b2 was 59.3% (95%CI: 40.7–77.9%) (Appendix A) and 49.7% (95%CI: 29.9–69.4%) against symptomatic infection in adolescents aged 12–18 years of age (Appendix A).

#### 3.4.3. Middle Aged Adults

One study reported VE (8%; 95%CI: 6.0–9.9%) of symptomatic infection in middle aged adults 40–64 (Appendix A) [25]. The pooled VE against symptomatic Omicron infection for a primary series plus one booster of mRNA vaccine or AstraZeneca vaccine dose was 47.4% (95%CI: 23.3–71.5%) (Appendix A).

#### 3.4.4. Older Adults

We observed a trend towards lower VE in the older age group (≥60 years of age). The pooled VE against any documented Omicron infection in older individual’s ≥ age 60 year was −6.9% (95%CI: −51.9–38.1%) following any two primary mRNA doses (BNT162b2 or mRNA-1273) (Appendix A). The pooled VE estimated for the first three months was 30.3 (95%CI: −10.0–70.7%) and substantially reduced to −58.2% (95%CI: −94.7–21.8%) within three to six months (Appendix A). There were not enough time data points to estimate VE longer than six months. Primary vaccination plus one booster dose of either the BNT162b2 or mRNA-1273 vaccines restored the protection to 57.9% (95%CI: 53.4–62.4%) within 3 months, and declined to 14.7% (95%CI: −22.0–51.5%) at six months (Appendix A). The pooled VE against severe infection following primary vaccine series was 58.5% (95%CI: 50.0–66.9%) (Appendix A) and increased to 81.6% (95%CI: 75.5–87.6%) following administration of one booster dose of mRNA or Vector vaccine (Appendix A). VE of the first booster dose against severe infection was 83.0% (95%CI: 76.0–90.0%) after ≥7 days and remained stable at 3 months (78.4%, 95%CI: 73.6–83.2%) (Appendix A). VE of the second booster dose against severe COVID-19 illness was estimated to be 86.8% as discussed above (Appendix A).

### 3.5. VE by Omicron Sub Linages

Only two studies reported VE for Omicron sub linages BA.1 [26] and BA.2 [21], whilst four studies [18,21,22,29] evaluated 26 different VE estimates against symptomatic Omicron infection by sub linages BA.1 and BA.2. Therefore, stratified meta-analysis on VE was only conducted for the latter. The primary series provided similar pooled VE against symptomatic Omicron BA.1 infection (11.3%, 95%CI: −11.3–33.8%) and BA.2 infection (15.1%, 95%CI: −10.2–40.3%) (Appendix A). VE of the first booster dose against symptomatic Omicron BA.1 was slightly higher for BA.1 (53.2%, 95%CI: 45.6–60.8%) compared to BA.2 (49.3%, 95%CI: 50.0–57.6%) (Appendix A). Pooled VE against severe BA.1 Omicron infection for the primary series plus one booster dose of any mRNA vaccine was 87.4% (95%CI: 70.5–104.4%), and for a primary series alone was 52.7% (95%CI: 29.4–76.0%). Similarly, pooled VE against severe BA.2 infection following primary series was 52.5% (95%CI: 27.5–77.6%) and increased to 87.3% following one booster of mRNA vaccination (Appendix A).

Additionally, we performed sensitivity analyses by study designs (cohort or case-control) and statistical methods employed to estimate the VE (Logit, Poisson, and Cox regression models). These subgroup analyses did not reveal any meaningful differences to the overall VE findings against all outcomes (noting small numbers in some subgroups) (Data not shown).

## 4. Discussion

Our meta-analysis of 28 studies, which included nearly 11 million individuals, provides evidence on VE and duration of protection of COVID-19 vaccines against SARS-CoV-2 Omicron infection and severe COVID-19. Our data suggest that primary vaccination series are not sufficiently protective against the Omicron infection and protection wanes substantially over time from 28% at three months to 4% at six months. Similar trends were observed for symptomatic Omicron infection following full vaccination, broadly consistent with recent review findings [14,40,41]. The waning of primary COVID-19 vaccination course was less pronounced against severe Omicron disease, decreasing from 64% at three months to 49% after six months, consistent with recent findings [14,40,41].

Our meta-analysis suggests that first booster dose restores and provides additional protection for all outcomes. VE of the first booster dose against any (51%) or symptomatic (57%) Omicron infection remained moderate for at least 3 months. Although there was limited data for longer follow-up, VE of the first booster dose against symptomatic Omicron infection waned to 33% at six months, falling below the WHO’s minimal criteria of 50% when considering the outcomes of infection and symptomatic disease. This suggests the waning effect is also present for booster vaccination, consistent with recent studies conducted during the Omicron-dominant period [42,43,44]. However, our review suggests that protection against severe Omicron cases remained robust up to 86% after a single dose of booster for at least up to six months, corroborating recent findings [14,40,41].

Vaccine waning following booster vaccination was lower among the younger age group compared to the older adults aged ≥60 years. VE of the second booster dose against symptomatic Omicron infection declined more rapidly in older adults aged ≥60 years from 58% at three months to 14% at six months [24]. However, high level protection against severe disease still remained up to 78% for at least six months after the second booster dose in older adults, who are more vulnerable to severe COVID-19 outcomes. As of 28 November 2022, 68.5% of the world population have not received booster doses [45]. Future research should continue to evaluate the VE of booster vaccination with longer follow-ups to determine the duration of protection against the Omicron variant.

This meta-analysis had several limitations. The included studies were highly heterogeneous in terms of study populations, statistical approaches employed, definitions of symptomatic or severe COVID-19 used, analysed time points after vaccination, and vaccination schedules and regimes. All these factors may contribute to the discrepancy in our VE estimates and limit the generalizability of our results. Although the included studies made some sort of adjustments to their final VE estimates, not all accounted for important confounders, such as previous SARS-CoV-2 infection, underlying comorbidities, socio-economic parameters and COVID mitigation strategies.

## 5. Conclusions

This meta-analysis from a wide variety of study types, settings and populations demonstrates that primary COVID-19 vaccination courses were limited in preventing infections and severe disease caused by the SARS CoV-2 Omicron variant. Our review highlights the importance of booster doses for protection against Omicron infection and, more importantly, in providing high levels of protection against severe Omicron disease, particularly among the elderly population. Further research on the real-world performance of the existing vaccines including the new Omicron-specific vaccines is needed.

## Figures and Tables

**Figure 1 vaccines-11-00224-f001:**
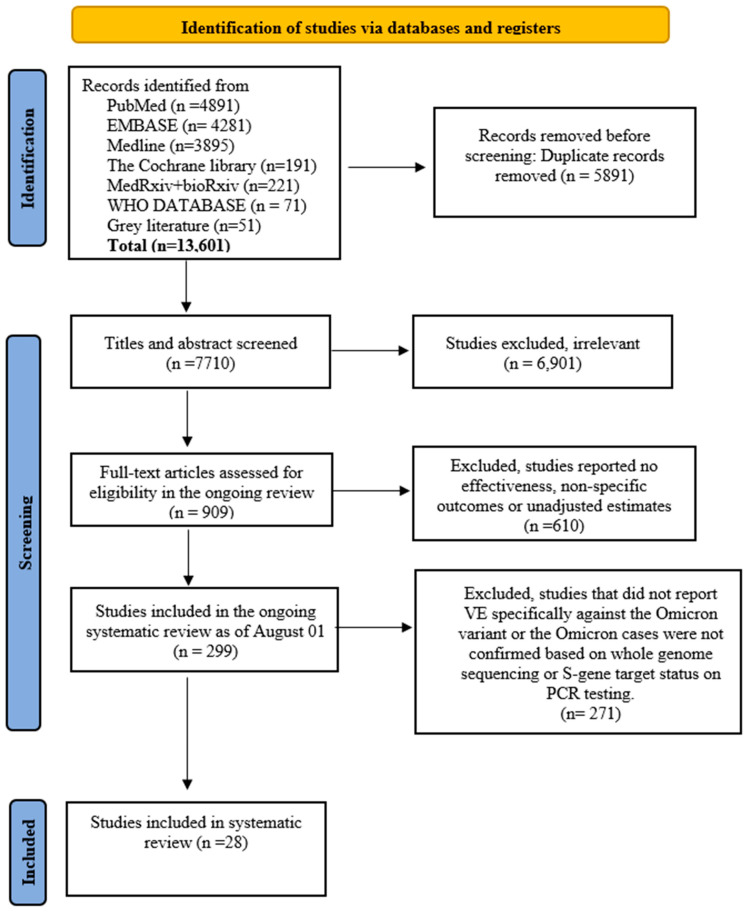
PRISMA flow diagram for study selection.

**Figure 2 vaccines-11-00224-f002:**
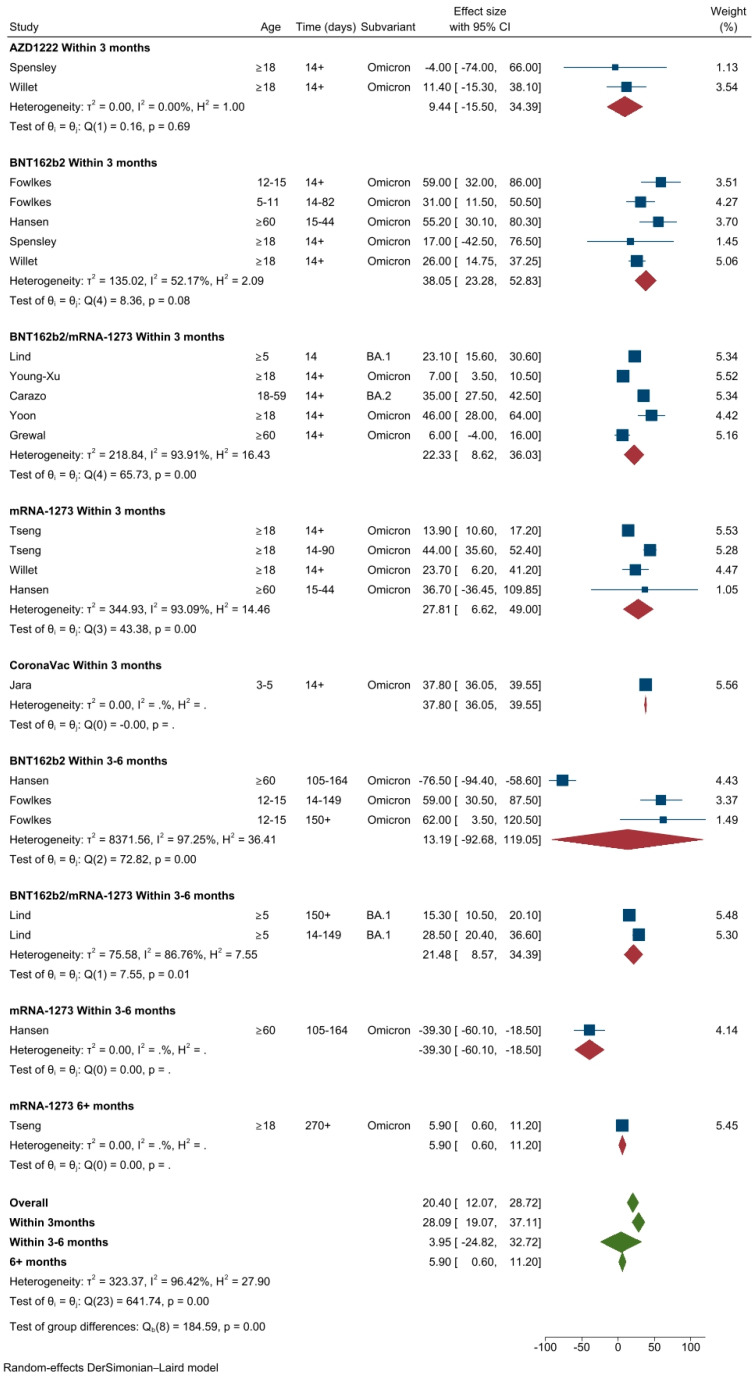
VE estimates against SARS-CoV-2 infection of the Omicron variant after the primary course, by vaccine types and time intervals.

**Figure 3 vaccines-11-00224-f003:**
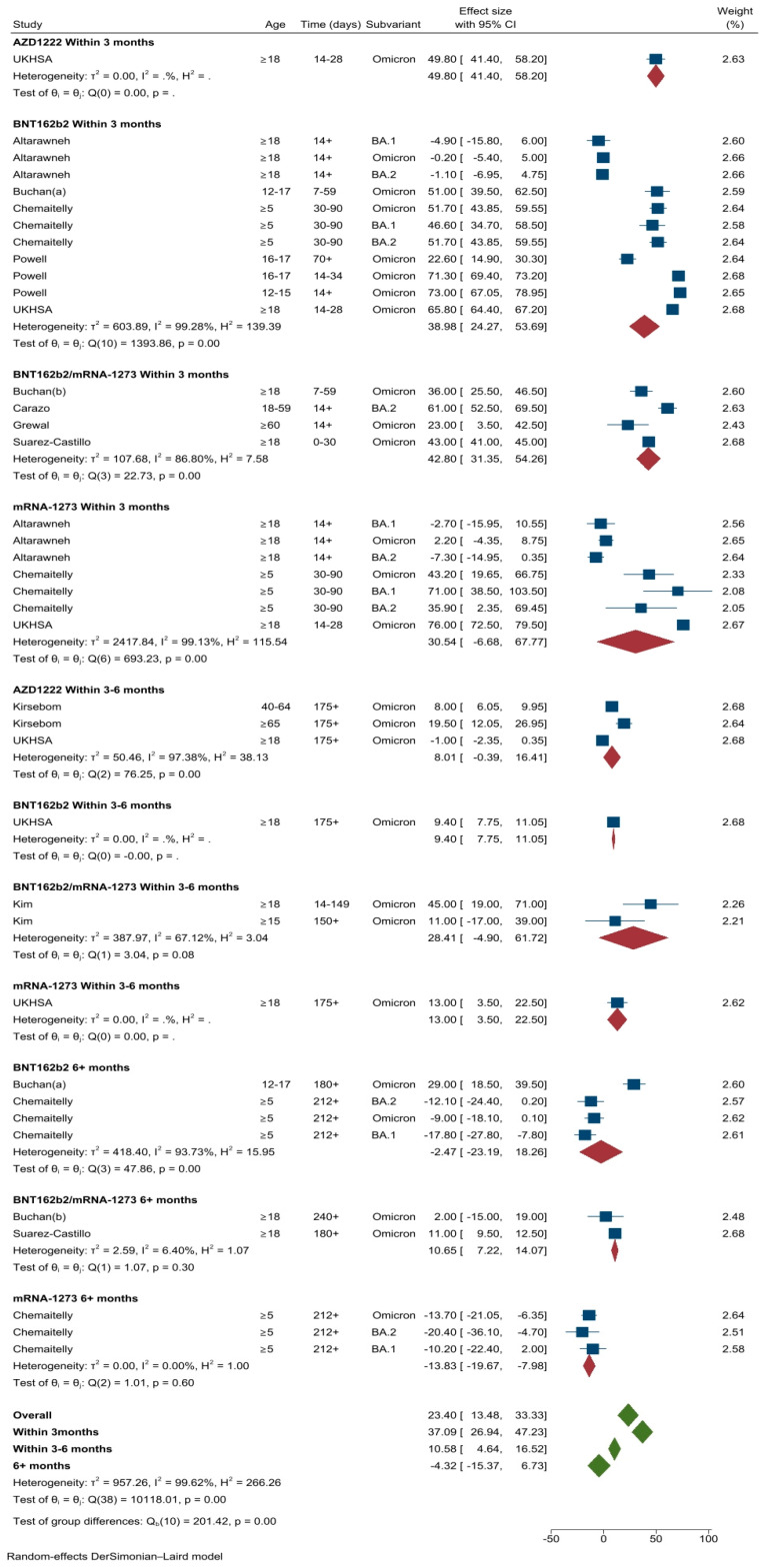
VE estimates against symptomatic Omicron infection after the primary course, by vaccine types and time intervals.

**Figure 4 vaccines-11-00224-f004:**
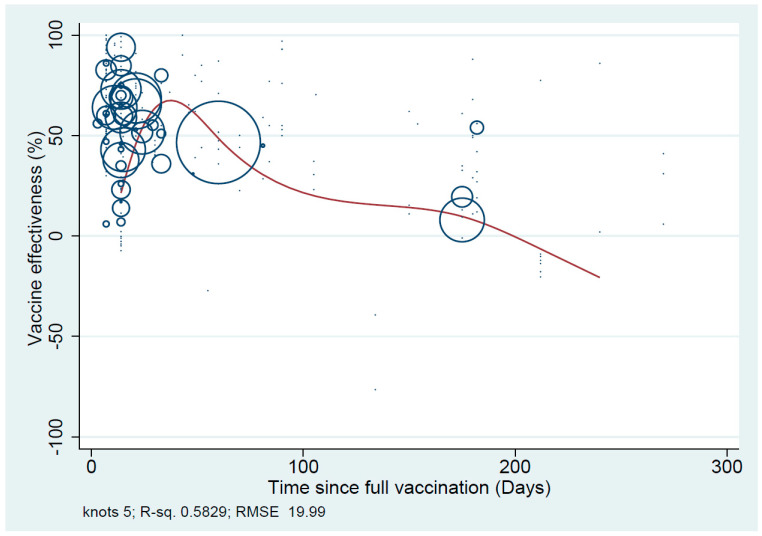
Scatterplot of VE against symptomatic Omicron infection plotted according to time from the primary vaccination.

**Figure 5 vaccines-11-00224-f005:**
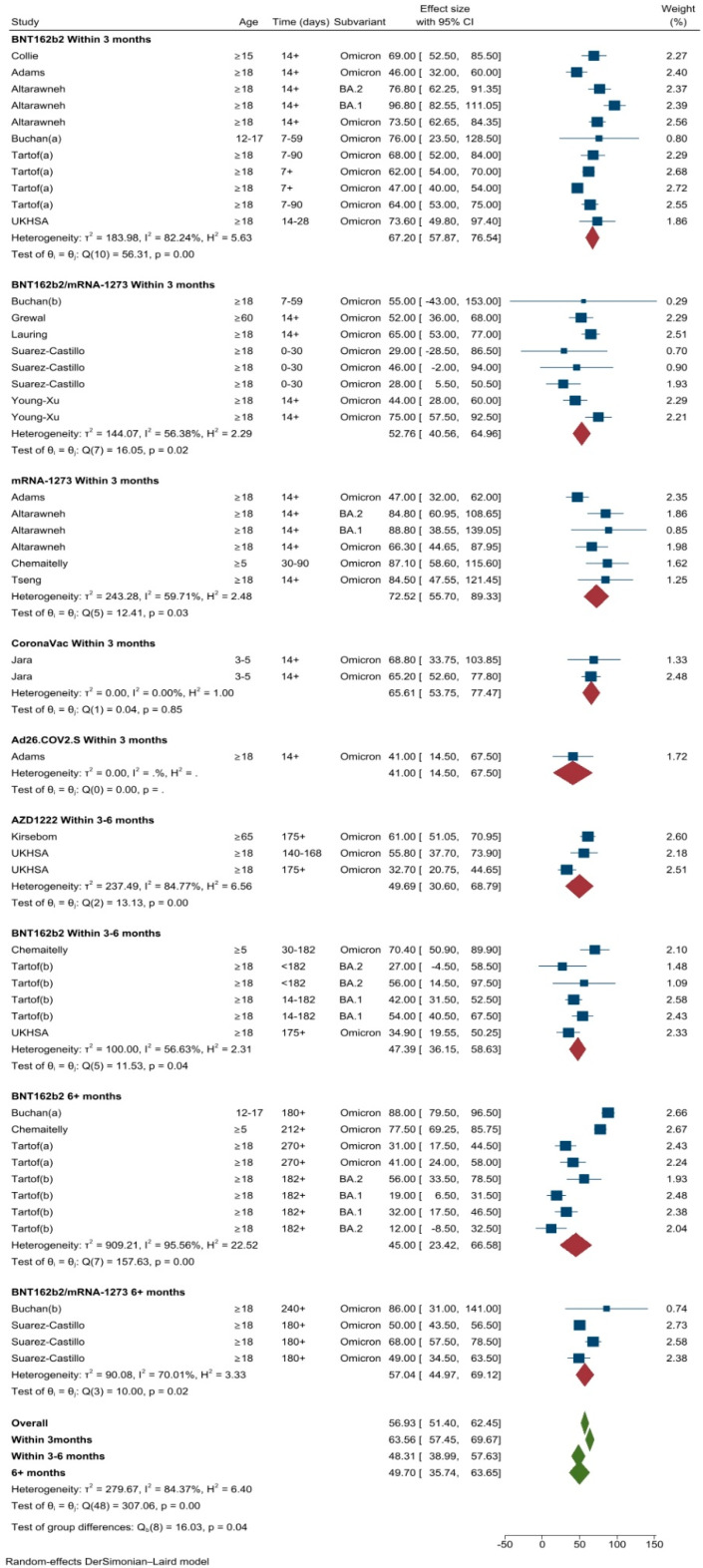
VE estimates against severe COVID-19 due to Omicron infection after the primary course by vaccine types and time intervals.

**Figure 6 vaccines-11-00224-f006:**
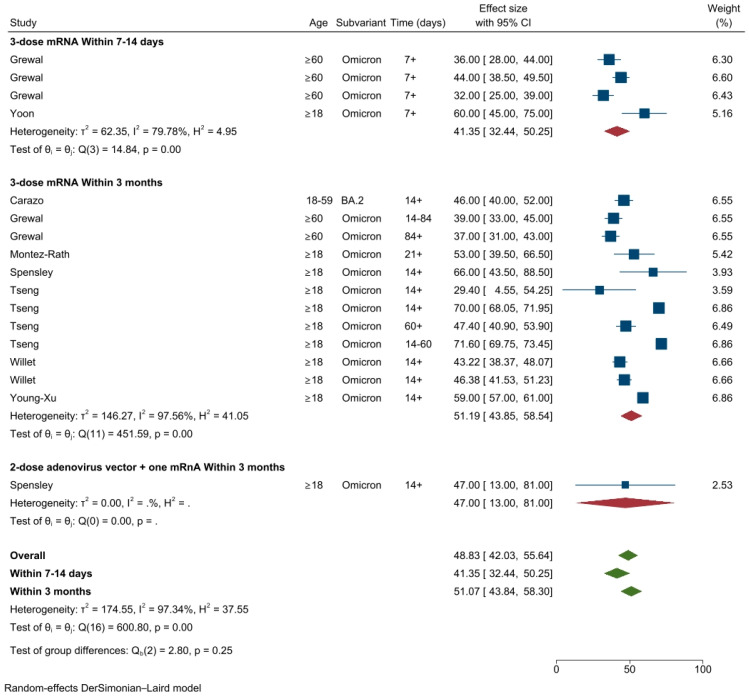
VE estimates against SARS-CoV-2 infection of the Omicron variant after one booster dose by vaccine types and time intervals.

**Figure 7 vaccines-11-00224-f007:**
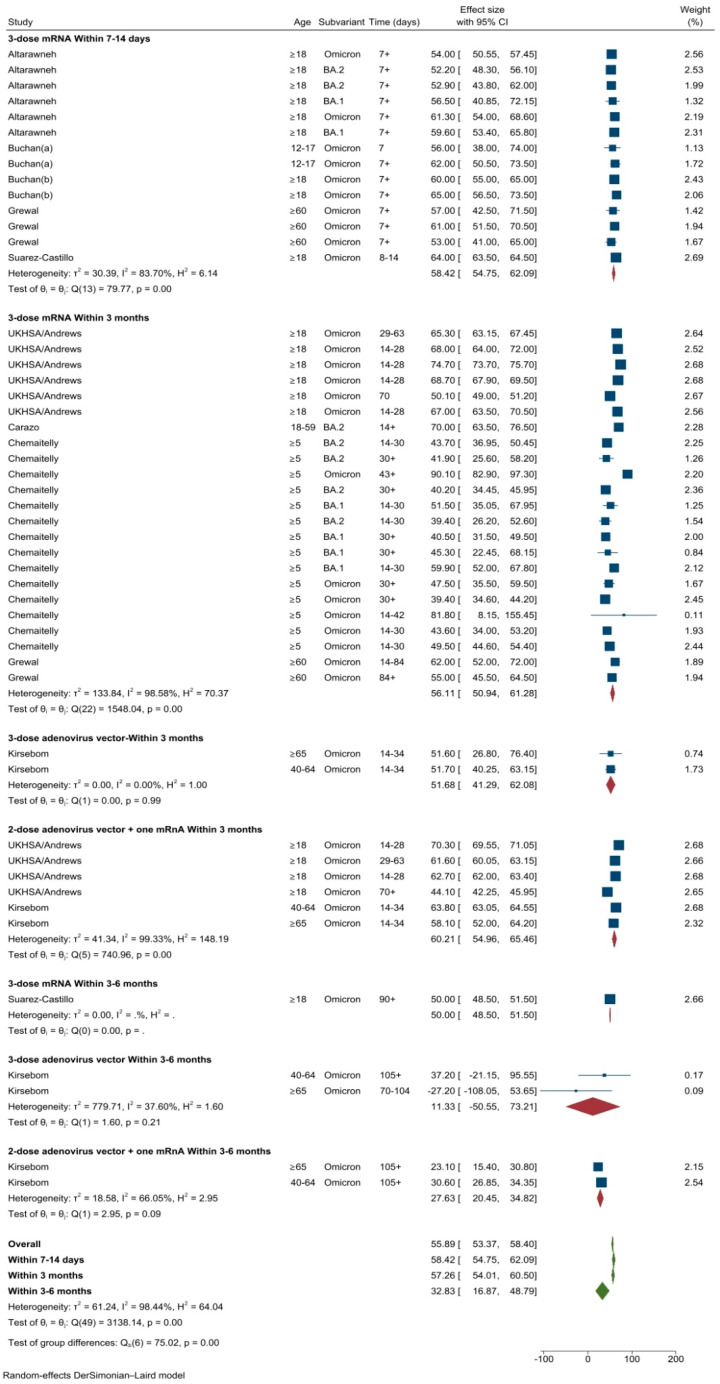
VE estimates against symptomatic Omicron infection after one booster dose by vaccine types and time intervals.

**Figure 8 vaccines-11-00224-f008:**
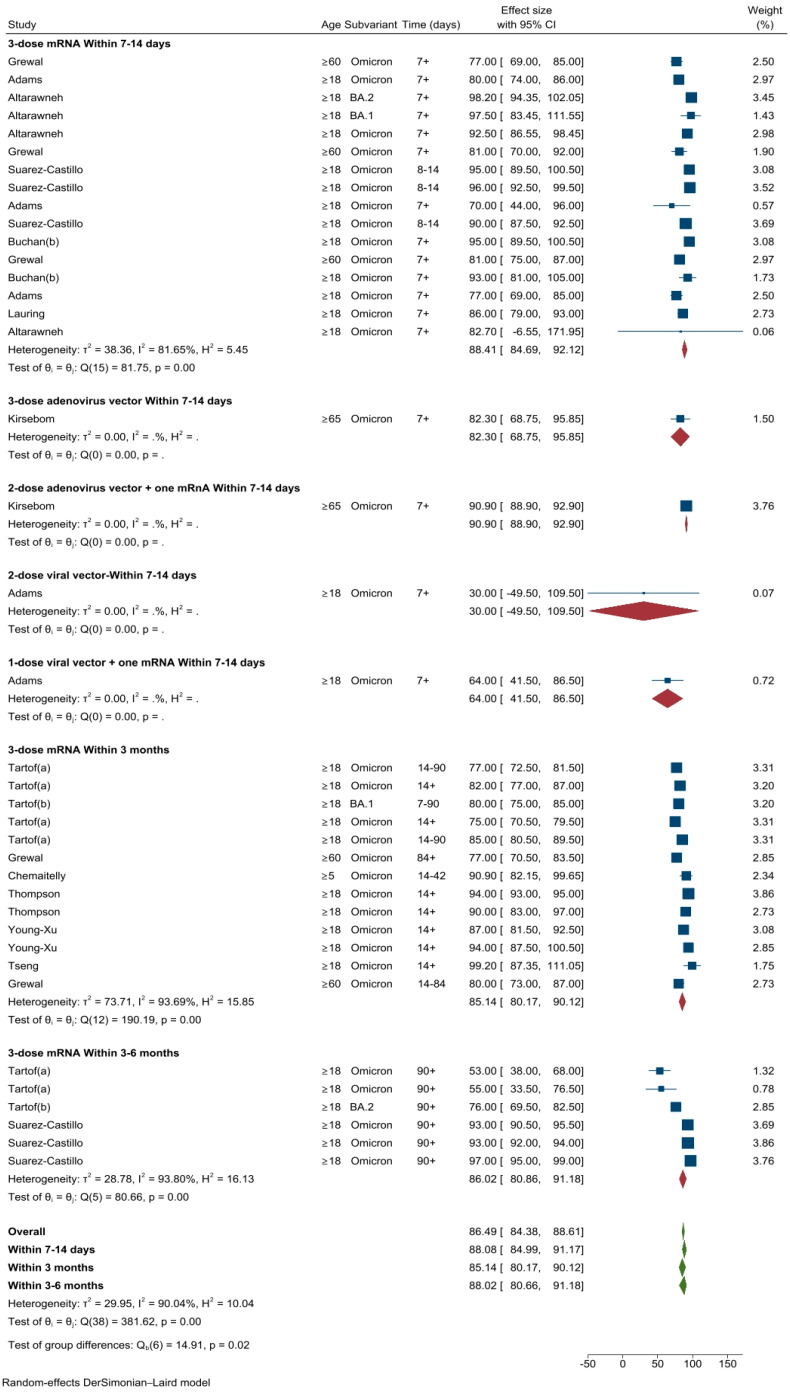
VE estimates against severe COVID-19 due to Omicron infection after one booster dose by vaccine types and time intervals.

**Table 1 vaccines-11-00224-t001:** Characteristics of the studies included in the meta-analysis.

Study	Country	Study Design	Population/Age Group	Sample Size	Type of Primary Vaccines	Time Interval since Primary Series (Days)	Type of Booster Vaccines	Time Interval since Booster Dose (Days)	Outcomes	VE/100%
Adams et al. 2022 [17]	USA	Case-negative control	Hospitalized adults ≥18 years	3181	Ad26.COV2.S, mRNA-1273, BNT162b2	14+	Ad26.COV2.S, mRNA-1273, BNT162b2	7+	Hospitalization	1-OR
Altarawneh et al. 2022 [18]	Qatar	Case-negative control	Individuals, all ages	158,484	mRNA-1273, BNT162b2	14+	mRNA-1273, BNT162b2	7+	Symptomatic infection, hospitalization/death	1-OR
Buchan et al. 2022a [20]	Canada	Case-negative control	Adolescents 12–17 years	29,855	BNT162b2	7–59, 180+	BNT162b2	7+	Symptomatic infection, severe infection	1-OR
Buchan et al. 2022b [19]	Canada	Case-negative control	General population ≥18 years	134,435	BNT162b2	7–59, 180+, 240+	BNT162b2, mRNA-1273	7+	Symptomatic infection, severe infection	1-OR
Carazo et al. 2022 [21]	Canada	Case-negative control	Healthcare workers aged 18–59	37,732	mRNA-1273, BNT162b2	14+	mRNA-1273, BNT162b2	14+	Documented infection, symptomatic infection	1-OR
Chemaitelly et al. 2022 [22]	Qatar	Case-negative control	General population ≥5 years	2,706,008	mRNA-1273, BNT162b2	30–90, 30–182, 212+	mRNA-1273, BNT162b2	14–30, 30+, 43+	Symptomatic infection, severe infection	1-OR
Collie et al. 2022 [23]	South Africa	Case-negative control	General population ≥5 years	211,610	BNT162b2	14+	NA	NA	Hospitalization	1-OR
Fowlkes et al. 2022 [36]	USA	Cohort	Children 5–11 years	1052	BNT162b2	14–149, 150+	NA	NA	Documented infection	1-OR
Grewal et al. 2022 [24]	Canada	Case-negative control	LTCF residents 60+ years	13,654	mRNA-1273, BNT162b2	14+	mRNA-1273, BNT162b2	7+, 14–84	Documented infection, symptomatic infection, hospitalization/death	1-OR
Hansen et al. 2021 [7]	Denmark	Cohort	Older adults 60+ years	41,684	mRNA-1273, BNT162b2	15–44, 105–164	NA	NA	Documented infection	1-HR
Jara et al. 2022 [8]	Chile	Cohort	Children 3–5 years	490,064	CoronaVac	14+	NA	NA	Documented infection, hospitalization, ICU admission	1-HR
Kim et al. 2022 [9]	USA	Case-negative control	General population ≥18 years	3847	mRNA-1273, BNT162b2	14–14, 150	NA	NA	Symptomatic infection	1-RR
Kirsebom et al. 2022 [25]	UK	Case-negative control	759,450 Middle-Aged Adults 40–64 & 759,450 Older adults 65+ years	166,720	AZD1222	175+	AZD1222, BNT162b2	7+, 14–34, 70–104, 105+	Symptomatic infection, Hospitalization	1-OR
Lauring et al. 2022 [26]	USA	Case-negative control	General population ≥18 years	17,126	mRNA-1273, BNT162b2	14+	mRNA-1273, BNT162b2	7+	Hospitalization	1-OR
Lind et al. 2022 [10]	USA	Case-negative control	General population ≥5 years	130,073	mRNA-1273, BNT162b2	14, 14–149, 150+	NA	NA	Documented infection	1-OR
Montez-Rath et al. 2022 [37]	USA	Cohort	Dialysis patients ≥18 years	3576	NA	NA	mRNA-1273, BNT162b2	21+	Documented infection	1-RR
Powell et al. 2022 [11]	UK	Case-negative control	Adolescents 12–17 years	617,259	BNT162b2	14–34, 70+	NA	NA	Symptomatic infection	1-OR
Spensley et al. 2022 [38]	UK	Cohort	Haemodialysis patients ≥18 years	1121	BNT162b2, AZD1222	14+	AZD1222, BNT162b2	14+	Documented infection	1-HR
Suarez Castillo et al. 2022 [27]	France	Case-negative control	General population ≥18 years	761,744	BNT162b2, AZD1222	0–30, 180+	BNT162b2, AZD1222	8–14, 90+	Documented infection, hospitalization, ICU admission, Death	1-OR
Tartof et al. 2022a [28]	USA	Case-negative control	General population ≥18 years	11,123	BNT162b2,	7–90	BNT162b2	14–90, 90+	Hospitalization, ED admissions	1-OR 1-HR
Tartof et al. 2022b [29]	USA	Case-negative control	General population ≥18 years	65,813	BNT162b2	14–182, 182+	BNT162b2	7–90, 90+	Hospitalization, ED admissions	1-OR
Thompson et al. 2022 [30]	USA	Case-negative control	Hospitalized adults ≥18 years	31,0676	Not stated	Not stated	mRNA-1273, BNT162b2	14+	Hospitalization, ED admissions	1-OR
Tseng et al. 2022 [31]	USA	Case-negative control	General population ≥18 years	109,662	mRNA-1273	14+, 14–90, 270+	mRNA-1273	14+, 14–60	Documented infection, hospitalization	1-OR
UKHSA 2022 [33]	UK	Case-negative control	General population ≥18 years	996,670	BNT162b2, AZD1222, mRNA-1273	14–28, 140–168, 175+	NA	NA	Symptomatic infection	1-HR
UKHSA/Andrews et al. 2022 [32]	UK	Case-negative control	General population ≥18 years	996,670	Not stated	Not stated	BNT162b2, AZD1222, mRNA-1273	14–28, 29–53, 70+	Symptomatic infection	1-IRR
Willet et al. 2022 [34]	UK	Case-negative control	General population ≥18 years	11,077	mRNA-1273, BNT162b2	14+	mRNA-1273, BNT162b2	14+	Documented infection	1-OR
Yoon et al. 2022 [39]	USA	Cohort	Healthcare workers ≥18 years	3241	mRNA-1273, BNT162b2	14+	mRNA-1273	7+	Documented infection	1-HR
Young-Xu et al. 2022 [35]	USA	Case-negative control	Veterans ≥18 years	372,636	mRNA-1273, BNT162b2	14+	mRNA-1273, BNT162b2c	14+	Documented infection, hospitalization, death	1-OR

## Data Availability

All data generated from the study have been included in the manuscript. Data were obtained from the primary studies included in the review.

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
