# Peer review of "A Systematic Review and Meta-Analysis on the Real-World Effectiveness of COVID-19 Vaccines against Infection, Symptomatic and Severe COVID-19 Disease Caused by the Omicron Variant (B.1.1.529)"

_vaccines, 2023, doi:10.3390/vaccines11020224_

Round 1

Reviewer 1 Report

The authors should be commended on preparing this detailed and well-researched systematic review. 

Specific comments: 

1. In the introduction section, it is relevant to mention that a recent social media analysis found that a topic of public interest was the public concerns’ regarding the new Delta and Omicron variants and the potential lack of efficacy of the ancestral COVID-19 vaccines against these new variants (citation: pubmed.ncbi.nlm.nih.gov/36146535). 

2. Information in the introduction section seems significantly dated. Suggest to update the information as many countries are way beyond BA.1 now. It was BA.2 around February to April this year, and then BA.4/5 and now many countries are being hit by XBB (which is a recombinant virus and progeny of BA.2.75 and BA.2.10). 

3. There is a recent meta-analysis that had essentially the same analyses and conclusions that primary vaccination does not provide sufficient protection against symptomatic omicron infection. (citation: pubmed.ncbi.nlm.nih.gov/36242513). This should be appropriately referenced and discussed in relation to the rationale for the present study. 

4. Authors may also want to highlight the genetic differences between omicron and the ancestral strains and emphasize how these mutations may affect immune evasion and the performance of vaccines. 

5. Please change "2. Materials and Methods" to "2. Methods". 

6. It should be explained that diagnosis of COVID-19 infections based on a positive lateral flow or rapid antigen test result was excluded given the higher rates of false positivity, especially for the omicron variant (citation: pubmed.ncbi.nlm.nih.gov/36431077). 

7. For Andrews et al. and Thompson et al. 2022, is it more accurate to say "not stated" for the primary series rather than "NA"? 

8. Please change "01 August 2022" to "1st August 2022". 

9. Please improve the image quality for the Figures, they are rather pixellated, especially for Figure 4. 

10. "These VE estimates varied based on vaccine type, for mRNA based; BNT162b2 38.1% (95%CI: 23.3%-52.8%), mRNA-173, 27.8% 172 (95%CI: 6.6-49.0%), BNT162b2 or mRNA-1273, 22.3 (95%CI: 8.6%-36.0 %) and for vector based; AZD1222, 9.4% (95%CI: -15.5%-34.4%) and CoronaVac 37.8% (95%CI: 36.1%-39.6%, data from a single study)" - this is quite misleading as the single study for CoronaVac consisted of young children aged 3-5 years and makes the comparison a very unfair one. 

11. Almost all the forest plots had very high heterogeneity >80%. How can we account for the very significant heterogeneity in the meta-analysis? Equally if not more important than the point estimate, the distribution of estimates (i.e., how much they are dispersed around the average pooled estimate) is a key aspect in meta-analyses. Are the studies truly comparable? 

12. Please change "vaccine schedules" to "vaccination schedules and regimes". 

Author Response

Reviewer #1

Comments and Suggestions for Authors. The authors should be commended on preparing this detailed and well-researched systematic review. 

Response: We thank reviewer#1 for their positive comment and careful review.

Specific comments: 

  1. In the introduction section, it is relevant to mention that a recent social media analysis found that a topic of public interest was the public concerns’ regarding the new Delta and Omicron variants and the potential lack of efficacy of the ancestral COVID-19 vaccines against these new variants (citation: pubmed.ncbi.nlm.nih.gov/36146535).

Response: Thank you for your comment. We have added this information to the introduction section on page 2, line 45-47 (tracked version).

“Recent social media analysis has shown increased public vaccine hesitancy due to the potential lack of effectiveness of ancestral COVID-19 vaccines against the new VOCs [10].”

  1. Information in the introduction section seems significantly dated. Suggest to update the information as many countries are way beyond BA.1 now. It was BA.2 around February to April this year, and then BA.4/5 and now many countries are being hit by XBB (which is a recombinant virus and progeny of BA.2.75 and BA.2.10).

Response: The following information has been added to the introduction section on page 1, line 36-41.

The omicron variant has rapidly evolved into new sub-lineages or sub-variants: BA.1 (B.1.1.529.1), BA.2 (B.1.1.529.2), BA.3 (B.1.1.529.3), BA.4 (B.1.1.529.4), and BA.5 (B.1.1.529.5). As of 29 November 2022, BA.5, BA.2.75, BA.4.6 and XBB (a hybrid of two different Omicron BA.2 sub-variants) are Omicron sub-lineages being monitored by the WHO to investigate if these lineages may pose an additional threat to global public health [5].”

  1. There is a recent meta-analysis that had essentially the same analyses and conclusions that primary vaccination does not provide sufficient protection against symptomatic omicron infection. (citation: pubmed.ncbi.nlm.nih.gov/36242513). This should be appropriately referenced and discussed in relation to the rationale for the present study.

Response:  The pre-print version of the meta-analysis (Meggiolaro et al.2022) has been cited in the discussion section in relation to the findings our review. The reference (published version) has now been updated. The systematic review (Meggiolaro et al.2022) was concluded on 1 March 2022 with 15 eligible studies conducted in the early Omicron era. Whilst our systematic review search ended on August 2022 and included 28 studies with longer follow-up period to evaluate the effectiveness of both primary COVID-19 vaccine course and booster doses. In addition, the meta-analysis (Meggiolaro et al.2022) does not include vaccine regimens created with inactivated vaccines such as CoronaVac.

We have added this information to the introduction section on page 2, line 54-59.

“A systematic review and meta-analysis has been recently published to evaluate the effectiveness of the current COVID-19 vaccines against Omicron infection [15]. This meta-analysis included 15 studies and demonstrated that primary vaccination does not provide sufficient protection against symptomatic Omicron infection [15]. However, the systematic review included vaccine effectiveness studies conducted in the early Omicron era with shorter-term follow up. The real-world long-term effectiveness and durability of protection conferred by primary COVID-19 vaccination series and booster doses against the Omicron variant is not precisely known.”

  1. Authors may also want to highlight the genetic differences between omicron and the ancestral strains and emphasize how these mutations may affect immune evasion and the performance of vaccines.

Response: The suggestion has been incorporated

Page 1 (line 33) “Compared to pre-Omicron variants, a number of mutations have been identified in the omicron variant, including multiple mutations in the receptor-binding domain of the spike protein associated with increased transmissibility and immune evasion after natural infection and vaccination [2, 3].”

Page 2 (line 45-46) “Several studies have shown diminished neutralization of Omicron by antibodies from vaccinated or convalescent individuals compared with previous SARS-CoV-2 variants [4, 11, 12].”

  1. Please change "2. Materials and Methods" to "2. Methods".

Response: Thank you for pointing this out. We have changed “Materials and Methods" to “Methods".

  1. It should be explained that diagnosis of COVID-19 infections based on a positive lateral flow or rapid antigen test result was excluded given the higher rates of false positivity, especially for the omicron variant (citation: pubmed.ncbi.nlm.nih.gov/36431077).

Response:   Our meta-analysis has only included studies with laboratory-confirmed SARS-CoV-2 Omicron infection based on whole-genome sequencing or S-gene Target Failure (SGTF) status using the RT-PCR. This has been mentioned in the study inclusion criteria on page 2, line 92-94 and on PRISMA flow diagram for study selection (Figure 1).

COVID-19 cases are defined as being due to the Omicron variant infection, based on S target–negative results on PCR or whole-genome sequencing.”

Figure 1- PRISMA flow diagram- Studies were excluded if Omicron cases were not confirmed based on whole genome sequencing or S-gene Target Failure (SGTF) on PCR testing.

  1. For Andrews et al. and Thompson et al. 2022, is it more accurate to say "not stated" for the primary series rather than "NA"?

Response: Thank you for this suggestion. This has now been amended.

  1. Please change "01 August 2022" to "1st August 2022".

Response: The date format has been changed.

  1. Please improve the image quality for the Figures, they are rather pixellated, especially for Figure 4.

Response: Image quality for the Figures has been improved.

  1. "These VE estimates varied based on vaccine type, for mRNA based; BNT162b2 38.1% (95%CI: 23.3%-52.8%), mRNA-173, 27.8% 172 (95%CI: 6.6-49.0%), BNT162b2 or mRNA-1273, 22.3 (95%CI: 8.6%-36.0 %) and for vector based; AZD1222, 9.4% (95%CI: -15.5%-34.4%) and CoronaVac 37.8% (95%CI: 36.1%-39.6%, data from a single study)" - this is quite misleading as the single study for CoronaVac consisted of young children aged 3-5 years and makes the comparison a very unfair one.

Response: The estimated VE for CoronaVac is now reported separately to avoid unfair comparison. The sentence has been modified, noting the VE estimates for CoronaVac is derived from a single study carried out in young children (Page 9, line 197-201).

“These VE estimates varied based on vaccine type, for mRNA based; BNT162b2 38.1% (95%CI: 23.3%-52.8%), mRNA-173, 27.8% (95%CI: 6.6-49.0%), BNT162b2 or mRNA-1273, 22.3 (95%CI: 8.6%-36.0 %) and for vector based; AZD1222, 9.4% (95%CI: -15.5%-34.4%). The estimated VE for CoronaVac was 37.8% (95% CI: 36.1%-39.6%, data from a single study) in children 3-5 years of age (Figure 2).

Almost all the forest plots had very high heterogeneity >80%. How can we account for the very significant heterogeneity in the meta-analysis? Equally if not more important than the point estimate, the distribution of estimates (i.e., how much they are dispersed around the average pooled estimate) is a key aspect in meta-analyses. Are the studies truly comparable? 

Response: We agree with the reviewer that almost the forest plots had high heterogeneity as mentioned in the limitation section. The high heterogeneity surrounding the primary meta-analysis estimates arises from the observational design of the included studies. To address this, we have generated forest plots of the pooled adjusted VE estimate from subgroup and sensitivity analyses. The subgroup and sensitivity analyses were performed by the time periods post vaccination, vaccine type /technology, age/population type and sub-lineages of Omicron. This enabled comparison of VE outcomes between similar subgroups across different studies. We also performed sensitivity analyses by study designs (cohort or case-control) and statistical methods employed to estimate the VE (Logit, Poisson, and Cox regression models). However, these sensitivity analyses didn't reveal any meaningful differences to the overall VE findings against all outcomes (noting small numbers in some subgroups) (Data not shown).

Page 21; line 350-354

“Additionally, we performed sensitivity analyses by study designs (cohort or case-control) and statistical methods employed to estimate the VE (Logit, Poisson, and Cox regression models). These subgroup analyses didn't reveal any meaningful differences to the overall VE findings against all outcomes (noting small numbers in some sub-groups) (Data not shown).

  1. Please change "vaccine schedules" to "vaccination schedules and regimes". 

Response: This has now been amended. (Page 21, line 371-372).

Reviewer 2 Report

I am most grateful for the opportunity of reviewing this SR on the effectiveness of COVID-19. As known, RWE is crucial to understand the real extent of protection offered by vaccination and to inform vaccination strategies and policies.

While it would be useful to have a comprehensive literature review on COVID-19 vaccines effectiveness, the presented work has some issues that make it not acceptable for publication.

Authors stated that “Outcomes of interest were VE against Omicron infection of “any type” (i.e. studies did not indicate underlying symptoms), “symptomatic COVID-19”, and “severe COVID-19” due to Omicron infection.”, but it is not clear whether authors included either studies with laboratory-confirmed Omicron-sustained infection or studies conducted during Omicron prevalence by time, place, country, subject.

It’s known that VE against COVID-19 is affected by several factors. The most relevant are vaccine type, mix-and-match of doses, viral circulation during the study period (that change the risk of being infected), behaviors (e.g., wearing mask that also change during vaccination period), previous infection, etc. Studies include in a SR/MA should be classified according to all these confounding factors to provide reliable pooled estimates of VE.

Quality assessment. While the Joanna Briggs Institute (JBI) tools has been widely used to assess risk bias of reports included in SR, for VE studies the ROBINS-I is a tool has shown superiority in evaluating risk of bias in the results of non-randomized studies that compare effects of vaccines, due the possibility of considering aspects of individual studies that are essential in VE assessment. For instance, quality also depends on the possible confounding factors listed in the previous point.  

Authors did not consider VE in prevention of death due to COVID-19. It is worth remembering that COVID-19 vaccines are primarily developed and licensed to prevent death and severe disease, while prevention of mild symptomatic disease or asymptomatic infection is an added value od vaccines.

Another point is related to pooling together studies that evaluate VE with different statistical analysis. There studies that “simply” considered OR, while others more correctly included time considering the time-varying effect of confounding factors listed above on the risk of breakthrough infection, severe disease, and death.

Author Response

Reviewer #2

I am most grateful for the opportunity of reviewing this SR on the effectiveness of COVID-19. As known, RWE is crucial to understand the real extent of protection offered by vaccination and to inform vaccination strategies and policies. While it would be useful to have a comprehensive literature review on COVID-19 vaccines effectiveness, the presented work has some issues that make it not acceptable for publication.

Response:  We would like to thank reviewer#2 for their suggestions.

Authors stated that “Outcomes of interest were VE against Omicron infection of “any type” (i.e. studies did not indicate underlying symptoms), “symptomatic COVID-19”, and “severe COVID-19” due to Omicron infection.”, but it is not clear whether authors included either studies with laboratory-confirmed Omicron-sustained infection or studies conducted during Omicron prevalence by time, place, country, subject.

Response: Our meta-analysis has only included studies with laboratory confirmed cases of SARS-CoV-2 Omicron variant infection based on whole-genome sequencing or S-gene Target Failure (SGTF) status using the RT-PCR. This has been mentioned in the study inclusion criteria on page 2, line 92-94 and on PRISMA flow diagram for study selection (Figure 1). Studies were excluded if Omicron cases were not confirmed based on whole genome sequencing or S-gene Target Failure (SGTF) on PCR testing (Figure 1).

COVID-19 cases are defined as being due to the Omicron variant infection, based on S target–negative results on PCR or whole-genome sequencing.”

It’s known that VE against COVID-19 is affected by several factors. The most relevant are vaccine type, mix-and-match of doses, viral circulation during the study period (that change the risk of being infected), behaviors (e.g., wearing mask that also change during vaccination period), previous infection, etc. Studies include in a SR/MA should be classified according to all these confounding factors to provide reliable pooled estimates of VE.

Response: Thank you for this suggestion. We have performed subgroup and sensitivity analyses by time intervals after vaccination, vaccine type /technology, age/population type and sub-lineages of Omicron. Most of the included studies were carried out during the omicron-dominant period and laboratory-confirmed cases of SARS-CoV-2 Omicron infection were only considered for inclusion in our meta-analysis. There was limited reporting of potential confounders such as history of prior SARS-CoV-2 infection and COVID-19-related health seeking behaviors (such as wearing mask or adherence to COVID-19 guidelines). This limitation has been mentioned on page 22-23, line 410-414

“Although the included studies made some sort of adjustments to their final VE estimates, not all accounted for important confounders, such as previous SARS-CoV-2 infection, underlying comorbidities, socio-economic parameters and COVID mitigation strategies’

Quality assessment. While the Joanna Briggs Institute (JBI) tools has been widely used to assess risk bias of reports included in SR, for VE studies the ROBINS-I is a tool has shown superiority in evaluating risk of bias in the results of non-randomized studies that compare effects of vaccines, due the possibility of considering aspects of individual studies that are essential in VE assessment. For instance, quality also depends on the possible confounding factors listed in the previous point.  

Response:  Both ROBINS-I and Joanna Briggs Institute (JBI) critical appraisal checklist are suitable tools to assess the risk of bias in non-randomized studies of interventions such as real-world effectiveness of vaccines. However, there is no consensus on the best tool to use for assessing the quality of observational epidemiological studies (Sanderson et al., 2007). ROBINS-I covers seven distinct domains through which bias might be introduced in observational studies. However, the tool provides limited guidance concerning key biases in the selection of cases and controls in case-control studies (Sterne et al., 2016). Specific versions of ROBINS-I for study designs other than cohort are under active development (University of BRISTOL). Overall, 22/28 of the included studies in our review are case-control studies. Therefore, JBI tools with 10 questions to guide the appraisal of case-control studies were preferred over ROBINS-I.

Authors did not consider VE in prevention of death due to COVID-19. It is worth remembering that COVID-19 vaccines are primarily developed and licensed to prevent death and severe disease, while prevention of mild symptomatic disease or asymptomatic infection is an added value of vaccines.

Response: Pooled VE estimates against death were not computed due to the low number of studies and insufficient data. Of the included 28 studies, only two (Suarez Castillo et al., 2022 & Young-Xu et al., 2022) evaluated the effectiveness of COVID-19 vaccines against death.  Altarawneh et al., 2022 & Grewal et al., 2022 included death or hospitalization as a composite outcome of severe COVID-19.  Therefore, we did not perform pooled VE estimates against death. 

This information is now added to the result section on page 14, line 255-258

“Only two [27, 35] studies evaluated the effectiveness of COVID-19 vaccines against death. Two studies [18, 24] included death or hospitalization as a composite outcome of severe Omicron infection. Therefore, VE estimates against death were not pooled separately due to the low number of studies and insufficient data.”

Response: We performed sensitivity analyses by statistical methods employed to estimate the VE (Logit, Poisson, and Cox regression models) to account for the difference in statistical approaches of the included studies. As mentioned in the result section on page 4, line 157-160, only 4.2% of the overall VE estimates examined vaccination as a time-varying covariate (time dependent hazard ratio (HR)) while the remaining VE estimates were derived from time-invariant analysis (OR, IRR, RR)

“Of the total 238 VE estimates, 196 (82.2%) were calculated from odds ratio (OR), 29 (12.1%) from the hazard ratio (HR), 10 (4.2%) from incidence rate ratio (IRR) and 3 (1.2%) from risk ratio (RR).”

We could not conduct pooled analyses for time-varying VE estimates or stratify by the vaccine type or time intervals due to small number of studies as mentioned on page 22; line 371-375

“Additionally, we performed sensitivity analyses by study designs (cohort or case-control) and statistical methods employed to estimate the VE (Logit, Poisson, and Cox regression models). These subgroup analyses didn't reveal any meaningful differences to the overall VE findings against all outcomes (noting small numbers in some sub-groups) (Data not shown).

Round 2

Reviewer 2 Report

The new version of the paper has substantially improved due to reviewers’ inputs. However, authors did not consider most relevant predisposing factors that could have an effect on COVID-19 vaccine effectiveness. I do not know the included primary studies what factors analyzed. But, this point should be included at least as narrative synthesis in Discussion. Possible predictors of vaccine response (VE) to be considered are: including age, smoking exposure, comorbidities and medication, type of vaccines, etc.

Author Response

Editor-in-Chief

Ms. Lynette Zhu

11 January 2023

Manuscript ID- vaccines-2105801

Title: A systematic review and meta-analysis on the real-world effectiveness of COVID-19 vaccines against infection, symptomatic and severe COVID-19 disease caused by the omicron variant (B.1.1.529)

Dear Lynette,

Thank you for the second round of Reviewers' comments and the opportunity to further revise the paper. We have included a point-by-point response to the reviewer’s comments outlined below.

Regards,

Hassen Mohammed

Research Associate,

School of Medicine, The University of Adelaide,

Vaccinology and Immunology Research Trials Unit,

Women's and Children's Hospital, North Adelaide, 5006, SA

T: +61 8 8161 9157

Email: hassen.mohammed@adelaide.edu.au

Reviewer #2

The new version of the paper has substantially improved due to reviewers’ inputs. However, authors did not consider most relevant predisposing factors that could have an effect on COVID-19 vaccine effectiveness. I do not know the included primary studies what factors analyzed. But, this point should be included at least as narrative synthesis in Discussion. Possible predictors of vaccine response (VE) to be considered are: including age, smoking exposure, comorbidities and medication, type of vaccines, etc.

Response: We would like to thank reviewer#2 for the suggestion.

A supplementary (Supplementary Material 3) table has now been added with list of covariates used in final analyses of vaccine effectiveness (VE) estimates from the included primary studies. The majority of included VE studies have accounted for key potential confounders that may influence both the receipt of COVID-19 vaccine and the occurrence of SARS-CoV-2 infection.

We have added this information to the result section on page 6, line 197-201.

“The majority of included studies have accounted for key potential confounding variables that influence both the receipt of COVID-19 vaccine and the occurrence of SARS-CoV-2 infection. List of covariates used in final analyses of vaccine effectiveness (VE) estimates from the included primary studies is reported in Supplementary material S3.”

Supplementary Material 3. List of covariates used in final analyses of vaccine effectiveness (VE) estimates from the included primary studies

Study

List of covariates used for matched case-control designs or adjustment in cohort studies

Adams et al.2022

Admission date (biweekly intervals), age, sex, race and ethnicity, U.S Health and Human Services region of admitting hospital

Altarawneh et al. 2022

Matched in a 1:5 ratio according to sex, 10-year age group, nationality, and calendar week of PCR testing in patients with the alpha, beta, and delta variants

Buchan et al 2022a

Age, sex, public health unit region of residence, comorbidities, influenza vaccination status during the 2019/2020 and/or 2020/2021 influenza seasons, positive test >90 days before index date, week of testing, and neighbourhood-level information on median household income, proportion of the working population employed as non-health essential workers, mean number of persons per dwelling, and proportion of the population who self-identify as a visible minority

Buchan et al 2022b

Age (in 10-year age bands), sex, public health unit region of residence, number of SARS-CoV-2 PCR tests during the 3 months prior to December 14, 2020 (as a proxy for healthcare worker status based on the start date of the provincial COVID-19 vaccine program), past SARS-CoV-2 infection >90 days prior to index date, comorbidities associated with increased risk of severe COVID-19, influenza vaccination status during the 2019/2020 and/or 2020/2021 influenza seasons (as a proxy for health behaviours), and neighbourhood-level information on median household income, proportion of the working population employed as non-health essential workers, mean number of persons per dwelling, and proportion of the population who self-identify as a visible minority

Carazo et al. 2022

Age (18–39, 40–59, and ≥60 years), sex, type of employment (as a proxy for socioeconomic status), facility (associated with infection risk and prioritization for vaccination), testing indication (as a proxy for disease severity), and epidemiological week (to address vaccine roll-out and potential differential in virus exposure opportunities over time between cases and controls)

Chemaitelly et al.2022

Cases and controls were matched two-to-one by sex, 10-year-age group, nationality, and calendar week of PCR test

Collie et al.2022

Age, sex, previous Covid-19 infection, surveillance week, geographic location, and the number of CDC risk factors.

Fowlkes et al.2022

Socio-demographic characteristics, health information, frequency of social contact, mask use, location, and local virus circulation

Grewal et al.2022

Age, sex, public health unit region of residence, week of test, whether they had tested positive for SARS-CoV-2 longer than 90 days ago, comorbidities, and whether there was an active SARS-CoV-2 outbreak in their long term care facility.

Hansen et al. 2021

Age, sex and geographical region, and calendar time as the underlying time scale.

Jara et al.2022

Age, sex, region of residence, nationality, health insurance category (a proxy of household income) and underlying conditions

Kim et al.2022

Sex, race/ethnicity, number of clinical encounters during 2019, number of underlying health conditions, and days since the previous infection

Kirsebom et al. 2022

Age (in 5-year bands, then everyone age 90 years or older), sex, index of multiple deprivations (decile), ethnic group, geographic region (NHS region), health and social care worker status, clinical risk group status (only available for those aged 64 years and younger, clinically extremely vulnerable (CEV) group status and severely immunosuppressed status

Lauring et al.2022

Calendar date of admission in biweekly intervals, US Department of Health and Human Services region (10 regions), age, sex, and self-reported race and Hispanic ethnicity

Lind et al.2022

Date of test, age, sex, race/ethnicity, Charlson comorbidity score, number of non-emergent visits in the year prior to the vaccine rollout in Connecticut, insurance status, municipality, and social venerability index (SVI) of residential zip code

Montez-Rath et al. 2022

Age, sex, and prior documented SARS-CoV-2 infection.

Powell et al.2022

Age, sex, index of multiple deprivation (quintile), ethnic group, geographic region (NHS region), period

(calendar week of onset), clinical risk group status (a separate flag for those aged over and under

16), clinically extremely vulnerable (if aged 16 and above) and previous positivity.

Spensley et al.2022

Age, sex, ethnicity, cause of kidney failure, previous transplant, immunosuppression at time of vaccine, diabetes, prior SARS-CoV2 infection

Suarez Castillo et al. 2022

Matching was based on age, sex , residence,  week of testing and presence of a comorbidity

Tartof et.al 2022a

Age, sex, race/ethnicity, Charlson comorbidity index, body mass index, and prior SARS-CoV-2 infection.

Tartof et.al 2022b

Age (18−49 years, 50−64 years, or ≥65 years), the month of emergency department or hospital admission, sex (male or female), race and ethnicity, body-mass index, Charlson Comorbidity Index, receipt of an influenza vaccine in the year before admission, receipt of a pneumococcal vaccine in the 5 years before admission (to adjust for health-care seeking behaviour), and documentation (PCR or lateral flow test) of previous SARS-CoV-2 infection (ever or never).

Thompson et al.2022

Age, geographic region, calendar time (days from August 26, 2021), and local virus circulation in the community and weighted for inverse propensity to be vaccinated or unvaccinated (calculated separately for each vaccine exposure group).

Tseng et al.2022

History of SARS-CoV-2 molecular test, preventive care, number of outpatient and virtual visits, Charlson comorbidity score, obesity (BMI ≥ 30), frailty index, specimen type, immunocompromised status and history of COVID-19. For the hospitalization models, the core variables included history of SARS-CoV-2 molecular test, preventive care, Charlson comorbidity score, obesity (BMI ≥ 30), immunocompromised status and history of COVID-19.

UKHSA 2022

Age (18 to 89 years in 5-year bands, then everyone ≥90 years), sex, index of multiple deprivation (quintile), race or ethnic group, history of foreign travel, geographic region, period (day of test), health and social care worker status, clinical risk-group status, status of being in a clinically extremely vulnerable group, and previously testing positive. These factors were all considered potential confounders and so were included in all models.

UKHSA/Andrews et al.2022

Age group, sex, ethnicity, self-reported vaccination status, geographical region of residence, and the week in which symptoms began

Willet et al.2022

Age, previous infection status, sex, SIMD quartile, and time since most recent vaccination.

Yoon et al. 2022

Propensity to be vaccinated,  influenza vaccination history, daily medication use, local virus circulation, study site and occupation

Young-Xu et al. 2022

Age, race, rurality, Veterans Health Administration (VHA) benefits priority and comorbid conditions (cancer, congestive heart failure, hypertension, immunocompromising conditions, obesity and diabetes).